# The Spectral Response of the Dual Microdisk Resonator Based on BaTiO_3_ Resistive Random Access Memory

**DOI:** 10.3390/mi13081175

**Published:** 2022-07-26

**Authors:** Ricky Wenkuei Chuang, Bo-Liang Liu, Cheng-Liang Huang

**Affiliations:** 1Institute of Microelectronics, Department of Electrical Engineering, National Cheng Kung University, Tainan 70101, Taiwan; a0981397292@gmail.com; 2Department of Electrical Engineering, National Cheng Kung University, Tainan 70101, Taiwan; huangcl@mail.ncku.edu.tw

**Keywords:** resistive random access memory (ReRAM), parallel double disk-coupled resonator, lithium niobite (LiNbO3)

## Abstract

With the resistive random access memory (ReRAM) devices based on the Al/BaTiO_3_ (BTO)/ITO structure fabricated at hand, by cross-analyzing the resistive memory characteristics in terms of various barium titanate (BTO) film thicknesses, it is found that the device with 60 nm thick BTO can be switched more than 425 times, while the corresponding SET/RESET voltage, the on-off ratio, and the retention time are −0.69 V/0.475 V, 10^2^, and more than 10^4^ seconds, respectively. Furthermore, the aforementioned ReRAM with a low switching voltage and low power consumption is further integrated with a waveguide resonator in the form of a dual microdisk aligned in a parallel fashion. As the separation gap between the two microdisks is fixed at 15 μm, the ReRAM-mediated dual disk resonator would render a 180° phase reversal between the spectral outputs of the through-port and drop-port. If the gap is shortened to 10 and 5 μm, the expected phase reversal could also be retrieved due to the selective combinations of different memory states associated with each of the two ReRAM microdisks as witnessed by a series of characterization measurements.

## 1. Introduction

The development of the compound semiconductor alloys and their particular use for realizing solid-state and semiconductor lasers, lay a concrete groundwork for modern optoelectronics and significantly increases the possibility of extending the electronic IC concept to photonics. The concept of photonic integrated circuits (PIC) was first brought to light by a special issue published in The Bell System Technical Journal in 1969 [1]. Since then, a plethora of PIC-related research manuscripts have been reported, many of which are primarily geared toward applications in optical communications [2,3,4,5,6,7]. On the other hand, the rapid development of electronic integrated circuits (IC) continues to overcome the limits of miniaturization. This so-called CMOS-compatible process is readily transferable to the makings of photonic integrated circuits (PIC), allowing many photonic components alike all to be packed on the same chip through either monolithic or heterogeneous means [8,9]. The integrations of silicon microelectronic and III-V compound semiconductor photonic components over the same platforms such as silicon substrate, the so-called optoelectronic integrated circuits (OEIC), deliver a wide variety of unique functionalities with unparallel advantages including immunity against electromagnetic interference (EMI), improved reliability, reduced chip areas, and comparably wider bandwidth compared to discrete circuits [10].

Out of these many prospective device components that could be potentially integrated on the same substrates, the memory, which could be volatile or nonvolatile [11,12,13] in nature, by far constitutes one of the indispensable members of OEICs that are heavily involved in signal processing and data computing. Generally speaking, both memory categories have their pros and cons. The volatile memory, even though it has a high storage and retrieval speed, however, the data retention can only be maintained while the electrical power is supplied, not to mention a high power consumption is usually needed to process every bit of data. On the other hand, nonvolatile memory data would remain intact for a very long, if not unlimited duration of time even when power is removed, albeit with comparably slower data access speed. The contemporary nonvolatile memory such as flash memory is generically based on the floating gate technique and is already commercially available [14]. To enhance its benchmarks, several other revolutionary new designs have already been reported, which include ferroelectric random access memory, or FeRAM [15], magnetoresistive random access memory, or MRAM [16], phase change memory (PCM) [17], and resistive random access memory, or ReRAM [18]. Among these four major products, ReRAM distinguishes itself mostly due to its relatively simple structure and ease of fabrication. It is in this regard that ReRAM shall be one of the targets considered for device integration.

If compared with other research topics, up until now there have not been an astronomical number of manuscripts reported or published that specifically concern the integration of optical waveguides with the memristors (memory resistors), which are mainly differentiated from one another based on the mechanisms upon which different memory states are switched. First of all, there is a phase change material (PCM) based memory switch that offers s high refractive index contrast between the crystalline and amorphous states. These PCMs can either be volatile such as vanadium oxide (VO_2_), which requires a continuous power supply to maintain a distinctive memory state [19], or nonvolatile such as Ge_2_Sb_2_Te_5_ (GST), which can maintain a state without perpetual power input [20,21,22]. The remaining others are also nonvolatile but with their memory toggling engineered through the resistive switching effect. This effect was first reported in a hybrid plasmonic waveguide with amorphous silicon utilized as an active material [23]. The previous concept has also been employed using silicon dioxide (SiO_2_) as an active layer [24]. The crux of integrating a resistive memory with an optical waveguide is to facilitate interaction between light and the conductive nanofilaments that can be formed or annihilated within the active layer through either or a combination of electrochemical metallization, valence change, and thermochemical mechanisms. However, the jury is still out when concerning which of the aforementioned mechanisms would play an influential role under any typical circumstances [25]. Electrochemical metallization relies on the diffusion from ions of a metal electrode, such as silver which is widely used in plasmonic memristors to facilitate resistive switching [23,26,27,28]. On the other hand, the valence change mechanism would have the upper hand on materials that are filled with oxygen-related defects or vacancies, which are instrumental for the formation of the conductive nanofilaments through ions migration. Consequently, oxygen ions migration also plays a conspicuous role in electro-optical memory switching as well [29,30,31,32]. Lastly, the thermochemical reactions and Joule heating may also impact the creation and destruction of the nanofilaments, which are primarily found in unipolar resistance switching (URS), or nonpolar resistance switching, although a substantial number of the resistive memristors that have been documented up to date are functionally bipolar [33,34,35].

Our waveguide-based resistive memristors, which have been published in the past [30,31], along with the present work, though not operating in plasmonic nature, do have one thing in common, that is, the devices are driven by the functionalities of electrical writability/erasability and optical readout. However, in dramatic contrast with other aforementioned works which have their active memory built directly on top of a passive waveguide, our novel proposal is contingent on the integration of two active cylindrical microresonators, which simultaneously behave as resistive memristors, with optical waveguides. In this paradigm, the very configurations can be readily scaled up to include more memory elements that be cascaded in either parallel or serial fashion, which could potentially be employed for complex logic operations. To briefly elucidate the inherent concept, with the two memory resonators currently under study and each disk being endowed with two memory states (1 and 0), we would have 2^2^ logic combinations. By increasing the number of devices that are possible for integration, the unique combinations go up to 2^n^! Therefore, it would be interesting to observe how would the spectral output be influenced by the unique memory state of each memory microresonator when operating in unison.

In this paper, Al/BaTiO_3_ (60 nm)/ITO resistive memory is to be integrated with bus waveguides as a dual disk waveguide microresonator. By being parallel, the resistive memory components in a shape of a cylindrical microdisk are arranged in a parallel fashion between the bus waveguides, which jointly constitute a two-port network. The resistive memory advantageously has a simple metal-insulator-metal (MIM) structure, low power consumption, scalability, and compatibility with the CMOS fabrication process, and is poised to become one of the next-generation nonvolatile memory. Here, each Al/BaTiO_3_/ITO ReRAM, or resistive memristor, adopted in our study, could achieve a combination of a low bipolar switching voltage, 425 switch cycles, a contrast ratio of 10^2^, and the data retention time of more than 10^4^ seconds at best. The ReRAM arranged in the form of the parallel dual disk microresonator, depicted in Figure 1, also optically functions as a unique spectral filter that depends on the relative phase shifts, which are dependent upon the respective high/low resistance states (HRS/LRS) of the two memristors. In a nutshell, our design paradigm graphically depicted in Figure 1 can be elucidated as follows. ReRAM typically comes with a metal-insulator-metal (MIM) structure, which, in turn, relies on toggling its memory state in a binary fashion between the high resistance and low resistance state, achievable through the breakage and formation of conductive filaments across the insulator between two electrodes. In this case, Al and ITO respectively serve as the bottom and top electrodes. These nanofilaments usually are predominantly made up of oxygen vacancies and in minutely small parts contributed by metal ions, as dictated by the current MIM structure. Therefore, with these memories manufactured in the form of cylindrical microdisks are simultaneously served as microdisk microresonators, we expect that the presence of the conductive nanofilaments would introduce an additional phase change to impact the overall spectral outputs. It is in this spirit that different memory states (HRS and LRS) of ReRAM can be correlated with distinctive spectral outputs. It is anticipated that the unique logic combinations would yield corresponding spectral fingerprints or signatures.

## 2. Device Fabrication

The microdisk memory fabrication procedures are schematically depicted in Figure 2. The lithium niobate substrate was first diced into several pieces before rinsing them sequentially with acetone, isopropanol, and DI water for 5 min at each cleaning stage. Next, titanium of 150 nm thick was electron-beam-deposited on LiNbO_3_ as a protective layer for undertaking the proton exchange process. After lithography and TMAH development, the titanium was etched by BOE to form a waveguide device pattern. Stearic acid was then used as a source for proton exchange conducted at a high temperature of 280 °C. Exchanging hydrogen ions furnished by stearic acid with lithium ions of lithium niobite creates an active region with an extraordinary refractive index ∆n_e_ of approximately 0.118 at the wavelength of 1552 nm.

After the exchange process, ITO of 200 nm thick was sputtered as the bottom electrode. Next, the area reserved for the memory pattern was defined and created by lithography, metal oxide sputtered deposition and lift-off. The metal oxide, barium titanate (BaTiO_3_) of 60 nm thick, was deposited by RF sputtering. Finally, aluminum of 150 nm thick as the top electrode was also defined and created by lithography followed by the sputtering and lift-off over BaTiO_3_. At this very stage, the fabrication of the optical memory is deemed complete and is ready for subsequent testings. The entire fabrication flow of the ReRAM-based dual disk waveguide resonator can be pictorially summarized in Figure 3 for clarity.

## 3. Results and Discussion

The integral parts of the waveguide resonator are the two memory disks placed in parallel between two bus waveguides. Before we proceed to characterize the resonator, it is imperative to evaluate the memory disks first to ensure their functionalities are in good order. The memory disks made up of Al/BaTiO_3_/ITO as mentioned previously are all tested by evaluating their I-V characteristics and both of them have demonstrated a bipolar hysteretic feature. As shown in Figure 4, one of the best memory devices could switch more than 425 times, while the corresponding SET/RESET voltage, the on-off ratio, and the retention time are −0.69 V/0.475 V, 10^2^, and more than 10^4^ seconds, respectively. The switching cyclic behavior of the memory is again shown in Figure 4, while the corresponding retention time is demonstrated in Figure 5.

To evaluate the barium titanate-based resistive memories that are integrated with dual optical waveguides in a parallel configuration, which all together make up a dual disk waveguide resonator, the input port is aligned with a 1550 nm fiber laser light source to facilitate the light input, and the power meter is used to ascertain the maximum light output could be obtained from the through-port and drop-port. After finalizing the optical alignment, the light source is replaced with a high-power EXFO T100-HP (EXFO Headquarters (Americas), QC, Canada) tunable laser, while the output power meter is swapped with the EXFO CT440 ((EXFO Headquarters (Americas), QC, Canada) optical component tester for subsequent spectrum characterization.

Without administering bias voltage to the memory device, the individual spectra taken from the through-port and drop-port of the ReRAM-based dual disk waveguide resonators in the wavelength range of 1530~1550 nm are gathered and compared to reveal the spectral differences as the gap distance between the two memory disks is varied (5, 10, and 15 μm). The corresponding results are presented in Figure 6, Figure 7 and Figure 8. It is found that a significant phase deviation between the two ports is detected when the two disks are separated by 15 μm. On the other hand, a barely noticeable phase shift is spotted when two disks are distanced by 5 and 10 μm. It is clearly shown that the separation distance between the two microdisks does affect the overall spectral shift, which in turn plays a vital role in influencing the transfer characteristics between the input and output optical fields.

The design intention of our ReRAM-based dual-disk waveguide resonators is as follows. According to Figure 1, as the light enters the input port, some of its energy could couple into disk 1 while the remaining energy would partly continue to traverse further to couple into disk 2 while the remaining portion heads straight to the through-port. Those light waves coupled into disks 1 and 2 would undergo countless circulations before being coupled out to the drop-port. The light waves traveling in the bus waveguides and within the disks would all experience phase delays; all these phase variations spectrally lead to constructive and destructive interferences, which eventually give rise to the overall spectral phase changes. Therefore, with the pertinent conditions per se, that is, the dimensions of the disks, the separation distance between the disks, and the distinctive memory states incurred during the device operation (HRS with filament breakage and LRS with filament linkage), all of these factors contribute to the overall phase changes.

As shown previously in Figure 1, each of the two ReRAM disks is numerically labeled for differentiating their biasing statuses. Three sets of dual disks respectively separated by 15, 10, and 5 μm are to be toggled between the high (HRS) and low (LRS) resistance memory states to record the phase changes of the light output accumulated at the through-port and drop-port. The required spectral data are extracted in response to the four different biasing conditions specified as follows. The first resistive memory (or numerically noted as ‘1’) is switched to LRS while the second resistive memory (or numerically noted as ‘2’) is not biased. Then, the ‘1’ is switched to HRS while keeping the ‘2’ unbiased. Next, once the ‘1’ is switched to HRS, the ‘2’ then is switched to LRS. Finally, the ‘1’ is switched to HRS while maintaining ‘2’ also in HRS. Therefore, different spectral data corresponding to the four different sets of biasing conditions are obtained for the dual disk waveguide resonators that have three different memory microdisk spacings (5, 10, and 15 µm). As the results demonstrated in Figure 9, Figure 10 and Figure 11, the rectangular areas denoted by the green and red boundary lines respectively depict the spectral windows showing 180° and zero phase shifts.

A series of spectra were first recorded from a dual disk resonator with two disks separated by 15 μm while maintaining each microdisk with a distinctive memory state. As shown in Figure 9a,b, whenever disk ‘1’ is switched into HRS or LRS while keeping disk ‘2’ unbiased, both spectra corresponding to the through-port and drop-port appear to be in phase. However, as shown in Figure 9c,d, when disk ‘1’ is maintained at HRS while disk ‘2’ is either in HRS or LRS, certain wavelength ranges (1530~1542.5 nm) of these two spectra with boundary line colored in green have demonstrated a complete phase reversal, while the ones within red blocks reveal only a slight spectral shift between the two ports.

As for the dual disk resonator with 10 μm spacing depicted in Figure 10, in contrast to Figure 7, a relatively narrow wavelength range over which the spectra corresponding to both ports have revealed a near, if not complete, phase reversal. As before, at this particular spacing, a similar mutual coupling between the two disks has also taken place as expected.

Similarly, a set of spectra corresponding to the waveguide resonator with two disks spaced by 5 μm is presented in Figure 11. As shown in Figure 11, only spectra contained within the wavelength blocks that are edge-colored in green, albeit narrow, have mani-fested a complete phase reversal, while ones in red blocks reveal only a relatively small shift between the two ports. We attribute the observed phenomena to the combined effects of different biasing conditions administered to each of two memory disks and much stronger coupling between the two disks as a result of their parallel configuration.

To summarize our observations based on the four different biasing conditions as elaborated earlier, whether or not a complete or near-complete 180° phase reversal has ever been achieved in each of the cases that involve different microdisk separations (5, 10, and 15 µm), all relevant findings are documented in Table 1. The results shown in Table 1 have two significant implications. First of all, different spectral contents strongly correlate with the memory state of each MIM microdisk (unbiased, LRS, or HRS), which helps to ascertain the memory states of these MIM disks via spectroscopic means. Second, as we scale up the number of memory microdisks aligned in parallel with one another, depending on the particular memory states (m) imposed on each of the n disks, an amazingly great number of different multi-bit logic combinations (m^n^) can be realized, which would have important applications in optical and neuromorphic computing in the future. 

## 4. Conclusions

Summarily, we have successfully integrated the Al/BaTiO_3_/ITO bipolar resistive memory with a low switching voltage into a configuration of a parallel dual microdisks memory resonator. Depending on the dual disk spacing and different memory states, namely, the high resistance (HRS) and low resistance (LRS) states imposed on each of the two memory disks, the different extents of the spectral phase shift running from barely noticeable to a complete reversal within certain distinct spectral windows are discovered between the through-port and drop-port. Most of all, our dual microdisk waveguide resonators have foreseeable applications including functioning as multi-bit logic elements in optical and neuromorphic computing.

## Figures and Tables

**Figure 1 micromachines-13-01175-f001:**
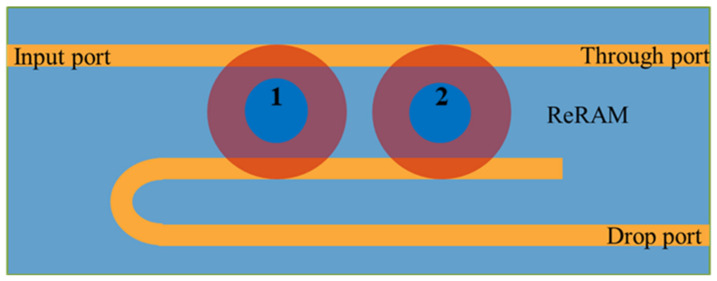
Plan view of the integrated ReRAM-based dual disk waveguide resonator. The numeric “1” and “2” refer to two ReRAM microdisks cascaded in a parallel configuration.

**Figure 2 micromachines-13-01175-f002:**
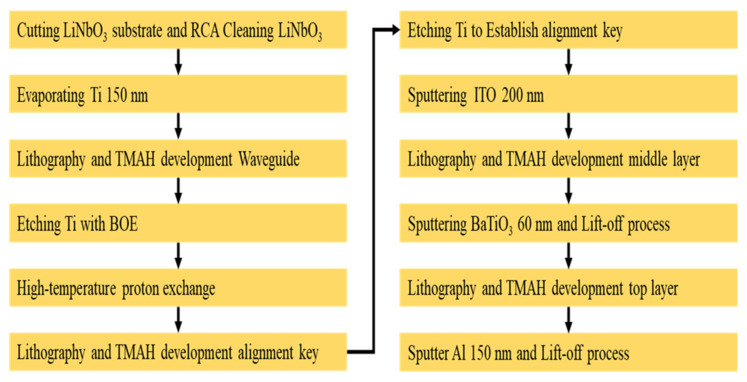
The fabrication process flow of a ReRAM microdisk.

**Figure 3 micromachines-13-01175-f003:**
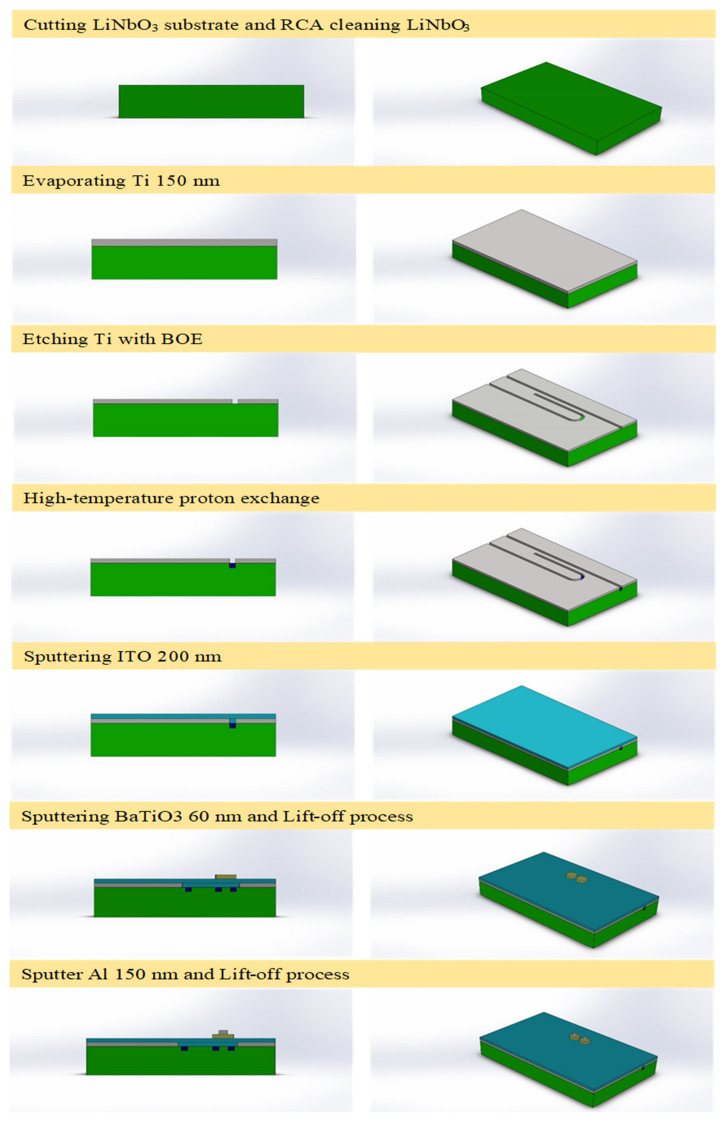
The fabrication process flow of a ReRAM microdisk-based waveguide resonator.

**Figure 4 micromachines-13-01175-f004:**
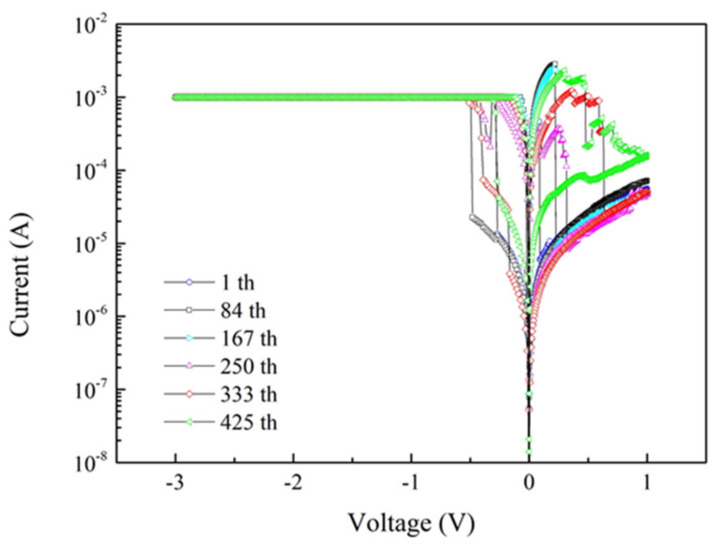
The I-V curves correspond to the maximum number of 425 cycles tested while each cycle shows a distinct set and reset voltages.

**Figure 5 micromachines-13-01175-f005:**
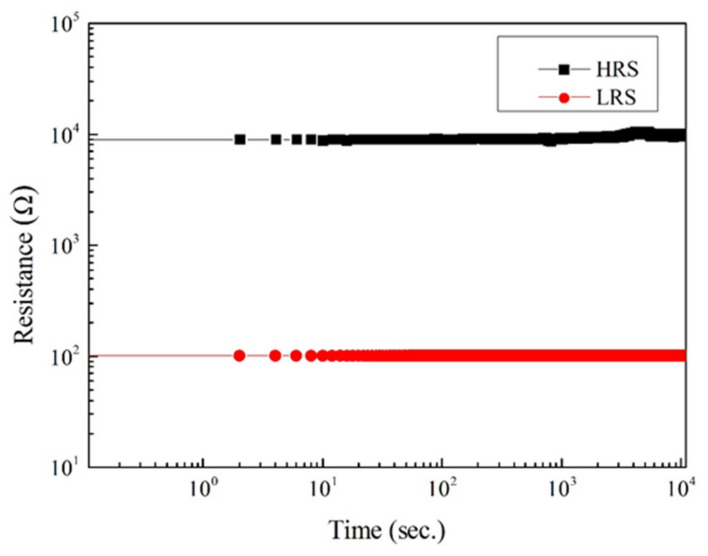
The retention time of Al/BaTiO_3_ (60 nm)/ITO resistive memory.

**Figure 6 micromachines-13-01175-f006:**
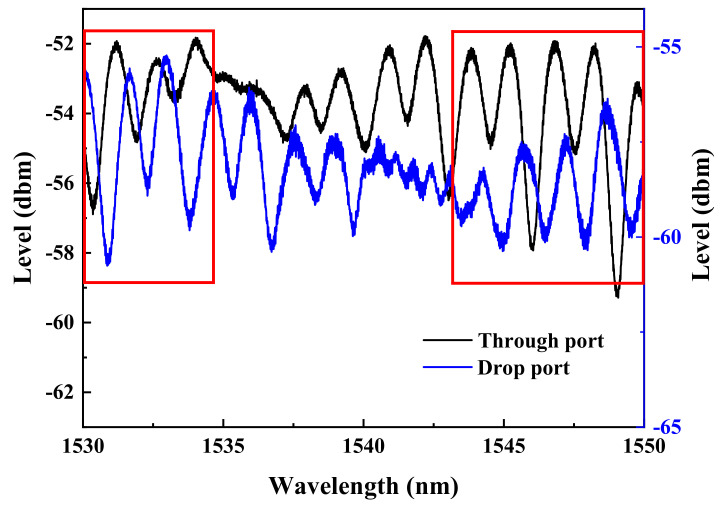
Through-port and drop-port spectra of the dual disk resonator with two microdisks separated by 15 μm.

**Figure 7 micromachines-13-01175-f007:**
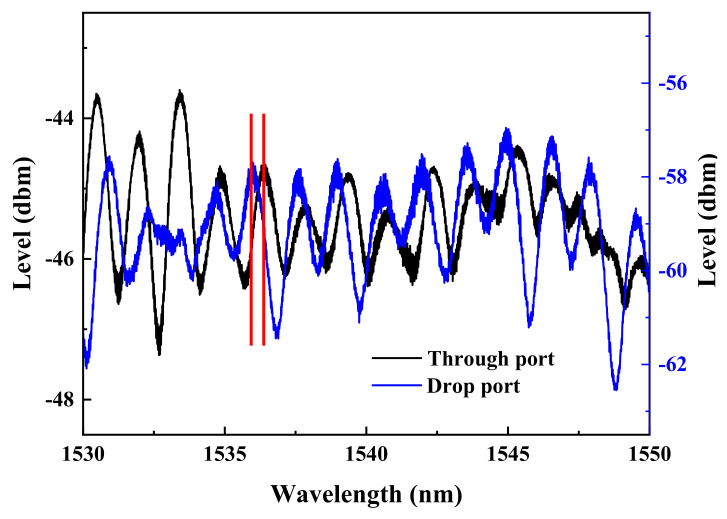
Through-port and drop-port spectra of the dual disk resonator with two microdisks distanced by 10 μm.

**Figure 8 micromachines-13-01175-f008:**
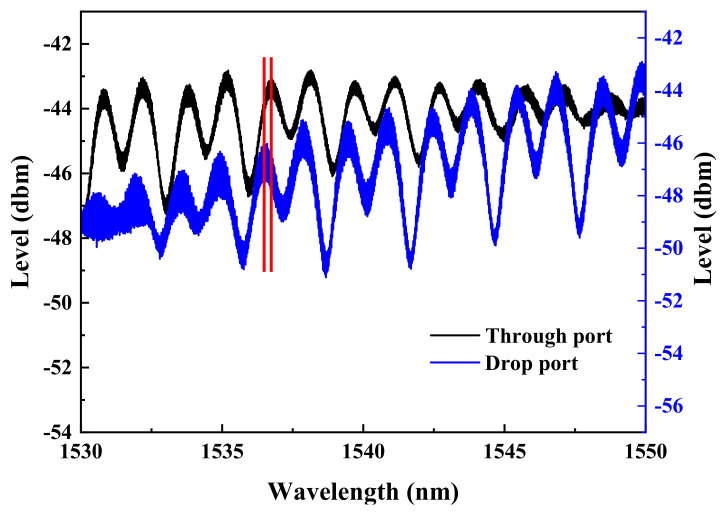
Through-port and drop-port spectra of the dual disk resonator with two microdisks were maintained at a gap length of 5 μm.

**Figure 9 micromachines-13-01175-f009:**
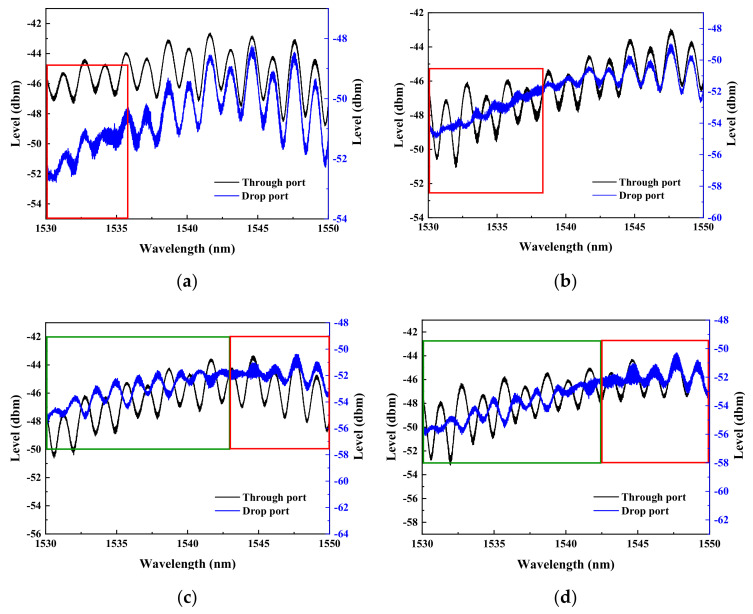
The spectra of the parallel dual disk resonator with the two disks separated by 15 μm are obtained when (**a**) disk ‘1’ is switched to LRS while disk ‘2’ remains unbiased, (**b**) the disk ‘1’ is switched to HRS while the disk ‘2’ stays unbiased, (**c**) the disk ‘1’ is set at HRS while the disk ‘2’ is biased to LRS, and (**d**) both disks ‘1’ and ‘2’ are biased to HRS.

**Figure 10 micromachines-13-01175-f010:**
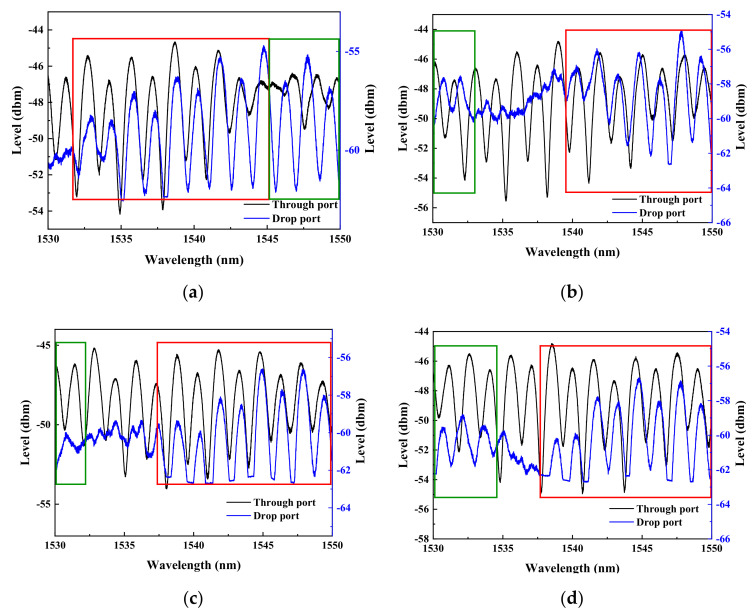
The spectra of the dual disk resonator with the two disks distanced by 10 μm are obtained when (**a**) disk ‘1’ is switched to LRS while disk ‘2’ remains unbiased, (**b**) the disk ‘1’ is switched to HRS while the disk ‘2’ stays unbiased, (**c**) the disk ‘1’ is set at HRS while the disk ‘2’ is biased to LRS, and (**d**) both disks ‘1’ and ‘2’ are biased to HRS.

**Figure 11 micromachines-13-01175-f011:**
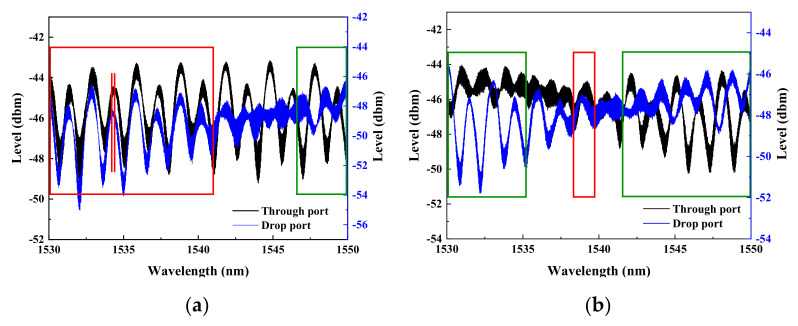
The spectra of the dual disk resonator with the two disks remaining separated at 5 μm are obtained when (**a**) disk ‘1’ is switched to LRS while disk ‘2’ remains unbiased, (**b**) the disk ‘1’ is switched to HRS while the disk ‘2’ stays unbiased, (**c**) the disk ‘1’ is set at HRS while the disk ‘2’ is biased to LRS, and (**d**) both disks ‘1’ and ‘2’ are biased to HRS.

**Table 1 micromachines-13-01175-t001:** Different biasing conditions are stipulated on each of the two memory disks to determine whether a near 180° phase reversal is achieved.

	15 µm	10 µm	5 µm
**‘1’ (unbiased)/‘2’ (unbiased)**	Yes	No	No
**‘1’ (LRS)/’2‘ (unbiased)**	No	Yes	Yes
**‘1’ (HRS)/’2‘ (unbiased)**	No	Yes	Yes
**‘1’ (HRS)/’2‘ (LRS)**	Yes	Yes	Yes
**‘1’ (HRS)/’2‘ (HRS)**	Yes	Yes	No

## Data Availability

Data is contained within the article.

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
