# Peer review of "The Spectral Response of the Dual Microdisk Resonator Based on BaTiO3 Resistive Random Access Memory"

_micromachines, 2022, doi:10.3390/mi13081175_

Round 1

Reviewer 1 Report

This paper proposed and fabricated the combination of ReRAM and resonator to realize the PIC. However, it needs major revision before publishing.

1. Figure 1 is not intuitive to infer how the device works. What is ReRAM and what is resonator? In ReRAM, which is Al electrode and which is ITO in the figure?

2. It would be better to update schematic illustration of process.

3. There are various active matrix material in the ReRAM area. Is there any reason to choose the BaTiO3 as an active matrix of ReRAM? Is BaTiO3 have play a role in the resonator?

4. If my understanding is correct, it is that ReRAM and resonators can be combined to represent multi-state. However, it is not clear how the resonator affects the conducting filament of ReRAM to express multi-state. It appears that the ReRAM and the resonator operate independently.

5. Where did you sweep the voltage and where on the ground in the ReRAM structure?

6. Can author confirm that the LRS/HRS of ReRAM changes every 15, 10, or 5 um gap of the resonator

Reviewer 2 Report

This paper reports the fabrication of dual microdisk resonator based on the BaTiO3 resistive random access memory, the basic characterization of RERAM performance, and the spectral phase shift taken from the through-port and drop-port of the ReRAM-based dual disk waveguide resonators in the wavelength range of 1530~1550 nm. In dependent on the dual disk spacing and different memory states (HRS or LRS) imposed on each of the two memory disks, the different extents of the spectral phase shift are discovered. These results suggest that a variety of multi-bit logic elements can be realized, depending on the memory states imposed on each of the disks.        

This article sufficiently novel and interesting to warrant publication. In the following, I raise a piece of comments to improve the paper before publication;

Comment 1: While table 1 summarizes the response of spectra in dependent on the dual disk spacing and different memory states, that is not fully explained. For example, why does the phase inversion appear or does not appear in a particular wavelength range and in a particular resistance state?

Comment 2: Remove the references 10-17.

Comment 3: Add (a)-(d) in Fig.7.

Comment 4: Add explanation about the green and red squares in the legends of Fig.7-9.

Round 2

Reviewer 1 Report

It would be OK to publish.

Author Response

Dear  Reviewer,

The submitted manuscript's introductory paragraph and the corresponding references have been expanded as suggested earlier by you and the editor. The entire manuscript has also been carefully proofread to remove grammatical ambiguities. Thank you!

Ricky Chuang

National Cheng Kung University